# The Effectiveness of the Sexual Reproduction in Selected Clonal and Nonclonal Species of the Genus *Ranunculus*

**DOI:** 10.3390/biology11010085

**Published:** 2022-01-06

**Authors:** Dawid Kocot, Ewa Sitek, Barbara Nowak, Anna Kołton, Alina Stachurska-Swakoń, Krystyna Towpasz

**Affiliations:** 1Department of Botany, Physiology and Plant Protection, Faculty of Biotechnology and Horticulture, University of Agriculture in Krakow, 29 Listopada 54, 31-425 Krakow, Poland; dawid.kocot@urk.edu.pl (D.K.); ewa.sitek@urk.edu.pl (E.S.); anna.kolton@urk.edu.pl (A.K.); 2Department of Plant Ecology, Institute of Botany, Faculty of Biology, Jagiellonian University, Gronostajowa 3, 30-387 Krakow, Poland; alina.stachurska-swakon@uj.edu.pl (A.S.-S.); krystyna.towpasz@uj.edu.pl (K.T.)

**Keywords:** apomictic species, fruit set, pollen viability, *Ranunculus illyricus*, reproductive biology, seed dormancy

## Abstract

**Simple Summary:**

The genus *Ranunculus* (buttercup) includes over 600 species, some of which are endangered, e.g., Illyrian Buttercup. Knowledge of the reproductive biology of such species may be crucial for conservation action. For this purpose, six species with different reproduction modes (nonclonal reproducing sexually by seeds only, clonal propagating by seeds and additionally vegetatively and apomictic) were observed. Selected features related to the efficiency of sexual reproduction were described: pollen viability, number of fruit set, seed viability and germination. It has been shown that in clonal species, which include the Illyrian Buttercup, the efficiency of sexual reproduction is lower compared to nonclonal species. The results will support conservation action taken for this species.

**Abstract:**

Generative processes have been evaluated in six European buttercup species in order to verify the hypothesis that the reproduction efficiency of clonal species is lower than that of nonclonal ones. The study covered common species (*Ficaria verna*, *Ranunculus auricomus*, *R. bulbosus*, *R. cassubicus*, *R. lanuginosus*) and the endangered *R. illyricus*. The following properties have been assessed: pollen viability (staining method), pollen grain germination and the pollen-tube elongation in pistil tissues (fluorescence microscopy), seed formation efficiency, seed viability (tetrazolium test) and germination ability by introducing factors interrupting dormancy (low temperature and gibberellin application). Additionally, the pistil morphology was documented for *R. bulbosus*, *R. illyricus* and *R. cassubicus* using SEM techniques. It was demonstrated that the reproductive efficiency, expressed as the production of viable seeds able to germinate, was significantly higher in the species reproducing sexually (especially in *R. lanuginosus*) compared to the clonal ones. However, the complexity observed leads to separation of an additional group (cluster) of apomictic species: *R. auricomus* and *R. cassubicus*, distinguished by the lowest pollen viability and a low ability of the seeds to germinate. In the vegetatively reproducing *R. illyricus*, the seed formation efficiency was just 13.2% despite the having highest number of pistils in its flowers. The developed seeds of this species observed in our experiment were viable, but in general effective methods to stimulate their germination have not been proposed yet. Here, the first comparative study concerning the biology of sexual reproduction of *R. illyricus* is presented in the context of its decreasing distribution in natural habitats.

## 1. Introduction

There are two modes of reproduction in plants: generative using seeds and vegetative (clonal) using bulbs, stolons or tubers. Numerous plant species are able to produce offspring both in vegetative and sexual ways, and the balance between the two reproductive modes may vary widely across and within the species. The problem of the trade-off between the resources allocated to vegetative versus generative reproduction within a plant has been investigated over recent decades [1,2,3]. The shares of sexual and clonal progeny may vary and depend on ecological or genetic factors that limit one or the other reproductive mode [4]. A question worth answering is to what extent the efficiency of sexual reproduction depends on the genetically determined reproduction mode.

An interesting choice for such investigations is the *Ranunculus* (buttercups) genus, which comprises about 600 species prevalent in both hemispheres, mainly in the subtropics and temperate zones [5]. Taking into account its taxonomic diversity, this genus is divided into 20 sections [6]. These are annual or perennial herbaceous plants growing in a variety of habitats, from damp meadows through forests up to xerothermic grasslands and mountain areas, as well as in streams and water reservoirs [5,7]. In the previous systematics, the genus *Ranunculus* also included representatives of the contemporary genus *Ficaria* [8,9].

Buttercups are characterized by different reproduction modes. They comprise species reproducing only generatively—self-fertilizing species or self-incompatible species—and those that have developed effective ways of vegetative reproduction [10,11,12]. There are species developing tubers in the axils of lower leaves (*Ficaria verna* Huds.), aboveground and underground stolons (*R. asiaticus* L., *R. cymbalaria* Pursch*, R. repens* L.), shoots rooting in the nodes (*R. flammula* L., *R. hederaceus* L., *R. reptans* L.), and other permanent underground storage organs (*R. asiaticus, R. illyricus* L., *R*. *nigrescens* Freyn, *F. verna*) [6]. Some species are optionally apomictic, asexually reproducing via seeds, for example, *R. auricomus* L. or *R. kuepferi* Greuter & Burdet [13,14,15,16,17]. Different strategies preventing self-pollination have been described in cross-pollinated species, such as protogyneous or protandrous flowers [11,18,19]. Both forms of temporal separation of gender phases have been described in *Ranunculus*, especially among the alpine species. Protogyny appeared to be more common and was described for alpine species of New Zealand, USA, lowland populations of *R. acris* L. and *R. repens* in the United Kingdom [18] and cultivated *R. asiaticus* [20]. Protandry of different degrees was reported for *Ficaria verna* [18] and New Zealand populations of *R. acris* and *R. flammula* [21].

The effectiveness of the sexual reproduction noted for *Ranunculus* species varies significantly and results from a proper course of successive stages: production of pollen, pollination, fertilization, seeds’ development, their viability and their ability to germinate. The course of these processes is affected by abiotic environmental factors, such as temperature [22,23,24], light [25] or water [26], as well as the presence of pollinators, which is the case for the self-compatible species *Ranunculus adoneus* A. Gray [22].

The regular course of generative processes depends not only on the availability of pollen, but also on its quality; for self-incompatible species, a precondition for seed formation is pollen of a different genotype [11] high viability. In the case of buttercups, studies have also looked at the size and viability of pollen grains in apomictic species [17,20,27,28,29]. Izmaiłow [16] conducted research on the apomictic complex of *R. auricomus,* and showed that a change in the plants ploidy level clearly reduced pollen viability and ability to germinate by ca. 30% and 75% in triploids and diploids, respectively.

For the reproductive success of a species, the formation of seeds that are alive and able to germinate is crucial. In many Ranunculaceae species at the time of diaspore dispersal, seed dormancy is determined by an undeveloped embryo or the structure of the seed coat [30,31,32]. Tiwari et al. [33] identified the dormancy of seeds in buttercups as an endogenous type resulting from underdevelopment of the embryo or physiological state. The natural ability of buttercup seeds to germinate varies across species, from 30% in *R. testiculatus* Crantz [34], to 50% in the case of *R. cortusifolius* Willd. [35], to even 96% for *R. peltatus* subsp. *baudotii* (Godron) Meikle ex C.D.K. Cook [30]. There are many ways in which the dormancy of seeds can be interrupted, and this effect has been studied for several buttercup species. Most commonly, seeds are exposed to low or high temperatures and growth regulators [30,34,35,36,37].

The aim of our research was to verify the hypothesis that the clonal species of *Ranunculus* exhibit limited effectiveness of generative reproduction (assessed based on pollen viability, number of pistils per flower, efficiency of fruit set, viability of seeds and ability of seeds to germinate) compared to nonclonal ones. Special attention was paid to *Ranunculus illyricus*—a rare species protected in some central European countries, the biology of which has not yet been studied and reported in detail.

## 2. Materials and Methods

### 2.1. Plant Material

Six species from the Ranunculaceae family, including five buttercups and one lesser celandine, were studied. The species come from a temperate climate, but from a range of habitats, and they differ in terms of their reproduction modes (Table 1 and Table 2). They are common in Central and Eastern Europe, with the exception of xerothermic *R. illyricus*, which is endangered in some countries in Europe including Poland [38,39,40].

Species reproducing only by seeds are regarded as nonclonal, while those that produces vegetative offspring using bulbs, tubers, rhizomes or stolons are called clonal [43]. Based on this criterion, our own observations (Table 2) and the literature, *R. lanuginosus* [6], *R. bulbosus* [12], *R. auricomus* [14,16] and *R. cassubicus* [44,45,46] were classified as nonclonal species.

The second group—clonal—was represented by *Ranunculus illyricus* and *Ficaria verna*, which are perennial geophytes producing underground clusters of tuberous roots. *Ficaria verna* was previously included in genus *Ranunculus* as *R. ficaria* [9]. In Central Europe, one can also come across the tetraploid *Ficaria verna* subsp. *bulbifera* Á. Löve & D. Löve, which additionally produces descendant tubers in leaf axils [9,47].

The test material comprised flowers and achenes collected from randomly selected individuals in their natural habitats (Table 2), except for *R. illyricus*, which was gathered from a collection at the Faculty of Biotechnology and Horticulture of the University of Agriculture in Kraków. Cultivated plants represented natural resources from one of the two known localities in Poland [39,48,49] and were grown in thermal, light, moisture and edaphic conditions in line with Ellenberg indicator values [50]. The research material was obtained at the optimum times for flowering and fruiting, which were different for each species (Table 2). Specimens in populations did not bloom synchronously, hence it was possible to collect at the same time flowers with open anthers, to assess the viability of pollen, and overblown flowers (flowers with a wilting corolla), to examine pollination effectiveness and pollen-tube elongation.

For every species, 30 and at least 50 (50–132) flowers or multiple fruits were collected in 2017 and 2019, respectively, each from a separate individual. The taxa chosen for the research form an apocarpous gynoecium, from which a spherical cluster of single seed fruits (achenes) develops. The effectiveness of pollination and fruit set were observed in the conditions of open—pollination of plants in natural stands (in situ)— or in the case of *R. illyricus*, an outdoor collection (ex situ).

Table 1 presents the systematic division of the genus into sections following Tutin et al. [6], the chromosome number and breeding system in accordance with PLADIAS [19], and the reproduction type according to Erikson [41], Sarukhán and Harper [12], Troll [42], and Tutin et al. [6].

### 2.2. Pollen Quality and Pistil Morphology

Anthers and pistils were excised from the flowers to assess pollen quality, germination and pistil morphology. Pollen viability was evaluated using the indirect staining method [51] on the collection day. The assessment was conducted in three repetitions. A single repetition involved a mixture of pollen collected from 10 flowers, each coming from a separate plant. In total, 300 grains of pollen were assessed in each repetition using a Zeiss Axio Imager M2 microscope (Carl Zeiss, Jena, Germany). Photographs were taken using an EOS 450D digital camera (CANON, Tokyo, Japan). At the same time, the diameters of pollen grains were measured using computer graphics program AxioVision 4.8 in three repetitions with 30 grains of pollen each.

Pistils from overblown flowers from the middle part of the receptacle were fixed in FAA (formalin, glacial acetic acid, ethyl alcohol 1:1:8 *v*/*v*/*v*) for 10–12 h, and then macerated in a 30% NaOH solution for two to three hours and cleared with a 6% H_2_O_2_ solution. Next, pistil tissues were stained with aniline blue for three hours [52] and squeezed between microscopic slides. The observations of pollination, pollen germination and the pollen-tube growth from the stigma to the ovary were conducted in fluorescent ultraviolet light with a wavelength of about 356 nm, using a Zeiss Axio Imager M2 microscope (Carl Zeiss, Jena, Germany). in the fluorescence mode. This allowed for an assessment of the percentage of pollinated pistils, and among them, the share of pistils with germinating and nongerminating pollen. There were at least 50 pistils per species analyzed.

For *R. bulbosus*, *R. illyricus* and *R. cassubicus*, additional documentation was prepared for pistil pollination and morphology using the SEM technique. The material was fixed in FAA, dehydrated in ethyl alcohol and dried in vacuum. The dried tissues were sputter-coated with gold and viewed under a Phenom™ ProX Desktop SEM electron microscope (ThermoFisher Scientific™, Waltham, MA, USA).

### 2.3. Efficiency of Seed Formation, Their Viability and Ability to Germinate

The efficiency of seed (single seed fruit) formation expressed as percentage was calculated as the number of ripe achenes per the number of pistils.

Seed viability was assessed using the tetrazolium method [53] in the year 2019. The achenes were soaked in water for 24 h, and then, after removing the pericarp, in a tetrazolium solution for the same period of time. The viability was evaluated for 15 to 58 seeds per species.

Seed material was also evaluated in terms of its ability to germinate using the blotter test in Petri dishes under a 16/8 h photoperiod and a temperature of 20 ± 2 °C [54]. The influence of factors interrupting seed dormancy was examined for a few variants: 4-week low-temperature stratification, pre-sowing conditioning with gibberellic acid (GA_3_), and a combination of both factors. Each treatment (and control) involved four Petri dishes (repetitions) with 25 seeds from each species.

In the stratification treatment, the seeds were kept at a low temperature (4 °C) for a period of four weeks. Application of gibberellin consisted of soaking seeds in a GA_3_ solution at a concentration of 1.0 × 10^−3^ mM for 24 h before sowing. Seeds that did not undergo this treatment were soaked in water for the same time and were used as control. The number of germinated seeds was evaluated after 4 weeks.

### 2.4. Statistical Analysis

All statistical analyses were performed with STATISTICA v. 13.3. Quantitative variables (efficiency of fruit set, germination of seeds per plate, number of pistils per flower, pollen diameter, pollen viability per sample) were tested for the normality of distribution and the homogeneity of variance. The normality of data in groups (species or clusters) was tested by a means of the Shapiro–Wilk test. The homogeneity of variance in groups (species or clusters) was tested by employing the Levene test. When comparing the groups with anormal distribution and homogeneity of variance, parametric tests (ANOVA) were used. However, when comparing the groups characterized by the lack of a normal distribution or the lack of homogeneity of variance, nonparametric tests (Kruskal–Wallis test) were applied. The null hypothesis tested stated that there were no differences between the groups (species or clusters), and the rejection of the null hypothesis allowed accepting the alternative hypothesis implying the existence of differences between the groups. Details of the tests used are provided in the captions of tables or figures. To assess the relationship between qualitative data, multiway contingency tables and the chi-squared test were used. The null hypothesis stated that there was no association between the group (species or cluster) and the parameter (viability of seeds). After rejection of the null hypothesis, it was possible to accept the alternative hypothesis that there was a relationship between the group and the parameter (viability of seeds). Additionally, cluster analyses were carried out to divide the species into groups of similar characteristics (Euclidean distance, Ward’s method). The significance level was α = 0.05.

## 3. Results

### 3.1. Pollen Quality and Pistil Morphology

The pollen of the species studied in our work differed in its viability, from 38% for *R. auricomus* to 91% for *R. bulbosus* (Table 3, Figure 1a–c). As far as size is concerned, pollen grains were similar to each other across the species (Table 3).

The SEM-aided observation revealed differences in the morphology of pistils across the species studied in our research (Figure 2). The differences concerned the pistil shape and the presence of ovary hairs: in *R**. illyricus*, the ovary was glabrous (Figure 2a,d), single long hairs were present at the base of the *R**. bulbosus* ovary (Figure 2b,e), whereas in *R*. *cassubicus*, a hairy ovary was observed (Figure 2c,f). Distinct differences were also noticed on the surface covered with papillae (stigma). A straight pistil of *R. illyricus* was covered with papillae up to about one-third of its height (Figure 2d). In the case of *R. bulbosus*, papillae concentrated only around the top part of the pistil (Figure 2e), while *R**. cassubicus* had papillae unilaterally covering the bent upper part of the pistil up to the ovary (Figure 2f).

Despite the different dates of flowering across the species, both in the case of the natural sites and in the collection, the pollination (in the open-field conditions) was effective because the majority of the observed stigmas were covered with pollen (Table 4, Figure 1d–i). There were numerous germinating pollen grains observed on stigma of all the species. For the majority of the species (not in *R. illyricus*), a single pollen tube entering the ovary was also seen.

### 3.2. Seeds’ Formation Efficiency, Their Viability and Ability to Germinate

The number of pistils per single flower varied across the species studied here (Table 3); extreme values were noted for two clonal species and ranged from 12 per flower for *F. verna* to 146 for *R. illyricus*. The species differed in terms of the effectiveness of fruit setting. The fruit set per flower estimated for *R. bulbosus* was 64%, while for *R. illyricus* it was only 11%. In this case, the clonal species were less effective than the other species, but also differed from each other. The highest viability of seeds was noted for *R. illyricus* and the lowest for *F. verna*. The species differed significantly in terms of the ability of seeds to germinate, with the best result observed for *R. lanuginosus* and *R. bulbosus* (Table 3, Figure 3). The species also responded in different ways to dormancy-breaking factors. 

Germination did not improve in the case of *F. verna* and *R. illyricus*, while for the germination of *R. auricomus* and *R. cassubicus* L., application of GA_3_ was crucial. GA_3_ also significantly improved germination of *R. lanuginosus* seeds. Low temperature decreased the germination ability of *R. bulbosus* (Figure 3). The combined action of the dormancy breakers did not improve germination parameters compared to the effect of either of the factors separately.

The species exhibited a variety of features, however, in order to decide if reproduction traits allow for distinguishing separate species groups, a cluster analysis was carried out based on pollen diameter and viability per sample, number of pistils per flower, efficiency of fruit set per flower, viability of seeds, and their ability to germinate per plate.

### 3.3. Cluster Analysis

Our cluster analysis formed four groups at 2.0 Euclidean distance. Cluster 1 with *R. bulbosus* and *R. lanuginosus*, taxa that reproduce only sexually, is characterized by the highest viability of pollen, highest ability of seeds to germinate, highest number of pistils per flower, highest efficiency of fruit set and high viability of seeds (Table 5, Figure 4). Cluster 2, comprising *R. auricomus* and *R. cassubicus,* is distinguished by the lowest pollen viability accompanied by a low ability of seeds to germinate and a medium number of pistils per flower. The clonal species belong to separate clusters. *R. illyricus*, which forms cluster 3, had the highest number of pistils per flower and the lowest efficiency of fruit set, and developed the most viable seeds, which germinated with difficulties. Cluster 4 with *F. verna* is characterized by high pollen viability, the lowest number of pistils per flower, and set seeds of low viability, which did not germinate.

## 4. Discussion

A comparative analysis of traits affecting sexual reproduction was carried out for five *Ranunculus* and one *Ficaria* species, out of which two are clonal perennials with two reproduction modes. The hypothesis to be verified assumed that the clonal species of *Ranunculus* have a limited effectiveness of generative reproduction compared to nonclonal ones. Our results confirmed this hypothesis, and additionally, the cluster analysis revealed that the relationships were more complex than expected. As a result, four groups of species (clusters) differing in their efficiency of sexual reproduction were separated.

The separation of clonal species from nonclonal species was confirmed, however, each of the clonal species constituted a separate group characterized by a low efficiency of generative reproduction expressed in the number of seeds capable of germination. Limitations to this efficiency appeared at different stages of generative reproduction: in *R. illyricus*, as a result of the low efficiency of fruit setting, and in *F. verna*, mainly following the formation of nonviable seeds. It was reported by Perje [55] that the low efficiency of sexual reproduction of *F. verna* was due to low viability of pollen unable to germinate on the stigmas. However, our investigation showed that pollen of this species was viable, germinated on the stigma and entered the ovary, in agreement with the report by Wcisło and Pogan [32].

In both clonal species, vegetative reproduction dominates, which relates to high allocation of resources to the production of vegetative propagules. Although these two species develop underground tubers, the mode of vegetative reproduction is different. Our observations of the *R. illyricus* plants in the collection revealed that the mother cluster of bulbs develops descendant clusters of bulbs (ramets) at the end of underground rhizomes for a few consecutive years. The underground clusters of tubers of *R. illyricus* are dedicated to new progeny, while the underground tubers of *F. verna* serve as a resource storage ensuring survival of the mother plant. Propagating bulbs of this species are formed on aboveground shoots in the axils of leaves.

Among the nonclonal species, two sister groups were distinguished: the first group comprised *R. lanuginosus* and *R. bulbosus*, whereas the second group included *R. auricomus* and *R. cassubicus*. This division is very interesting, because among the nonclonal species, a group of apomictic species has been separated [14,16] which were characterized by the lowest pollen viability accompanied by a low ability of seeds to germinate and a medium number of pistils per flower.

Our cluster analysis suggests that apomictic species are closer to the group of clonal species than sexual ones. The common feature of apomictic and clonal species is uniparental offspring developing without fertilization, copying favorable genotypes. However, in the case of apomixis, this process takes less input energy: resource allocation is similar to that of sexually reproducing species with genetically diversified progeny. Although facultative apomixis restricts recombination and reduces the evolutionary potential of the species, it maintains the fertility of individuals and is advantageous for colonization [56]. Apomixis could also support the fitness of a population under unfavorable environmental conditions [57].

Interestingly, the four clusters separated here coincide with the sections described within the genus *Ranunculus* by Tutin et al. [6] (Table 1) based on morphological traits. Similarly, studies [58] on the morphological and genetic diversity of apomictic and sexual *Cenchrus* species have shown that the apomictic species were clustered together separately from the sexual species. Moreover, the genetic distinctiveness was also evident from the assessed morphological features.

### 4.1. Seed Germination

The seed germination behavior is one of crucial traits in the life history of the plant species. According to the data available, different species of buttercups have different requirements for treatments breaking dormancy and stimulating germination [30,34,35,36]. It was shown [37] that the use of a low temperature is the most effective way to interrupt dormancy in species from the Ranunculaceae family growing in temperate climate and mountain areas. We have presented two groups of species that were distinguished in terms of the ability of seeds to germinate and their reaction to the applied factors breaking dormancy (Figure 3). One group included clonal and nonclonal facultative apomictic species, the seeds of which germinated worse than the seeds of sexual species. Neither GA_3_ nor low-temperature treatments improved the germination abilities of the clonal species, while gibberellin significantly increased the germination ability of the apomictic species.

Gibberellin, a natural phytohormone, is known for its ability to reduce seed dormancy time and successfully stimulate the seed germination of other plants. Gibberellin is an antagonist of abscisic acid, and both hormones are crucial for germination control [59,60,61]. Their proportions, and hence the seed germination ability, may be considerably modified by external factors, such as temperature [62]. Consequently, low- or high-temperature treatments are often used to stimulate germination [63]. Low temperature is the most typical factor controlling the physiological dormancy of seeds of plants in temperate climates [60]. Prechilling has been shown on numerous occasions to be effective in breaking seed dormancy for such species [64,65,66].

The seeds of the remaining nonclonal species in our experiment were characterized by a much greater ability to germinate, although the response of these species to the factors used was different. *R. lanuginosus* reacted like apomictic species, and GA_3_ was beneficial for seed germination. The reaction of xerothermic *R. bulbosus* seeds to stratification, which negatively affected the ability to germinate, was thought-provoking. Most likely, the plant was brought into secondary dormancy due to low temperature, a phenomenon already described for annual winter [67], some xerothermic [68,69] or desert species [70]. This mechanism may be important in preventing early germination of seeds in summer or early fall, when the conditions are not suitable for the development of new plant generations.

### 4.2. Pollen-Tube Development

Microscopic observations confirmed that there had been no restrictions on pollination both in the flowers from natural sites and from the collection (Figure 1d–i). Most of the pistils were pollinated in all the species under investigation and pollen germinated on stigmas profusely.

In all the species except *Ranunculus illyricus*, there were bundles of pollen tubes growing from the stigma towards the ovary where their growth was inhibited. Only for *R. bulbosus* was it possible to record a pollen tube entering the ovule (Table 4). In *Ranunculus*, which possess one ovule per carpel, a single pollen tube is sufficient for fertilization and fruit set. Our results corroborate those of Rendle and Murray [11] and Beruto et al. [20], who in the case of for *Ranunculus* also described large numbers of pollen grains capable of germinating, yet the growth of all but one pollen tube ceased and only that one tube reached the micropyle.

In the case of clonal *Ficaria verna*, for more than half of the specimens analyzed, numerous pollen tubes entering the ovary were observed. This contrasts with the results of Wcisło and Pogan [32] for this species, who reported only a few cases of single pollen tubes entering the ovary and only 7% of fruit set efficiency. The authors stated that the low efficiency of fruit set was a result of retarded growth of pollen tubes, and in consequence, a lack of fertilization in the majority of ovules. Our observations revealed a normal elongation of pollen tubes of *F. verna* corresponding to 25% of fruit set, but seed viability was only 7%. This means there must have been disturbances during the embryo development.

Different results were obtained for *Ranunculus illyricus*, in which numerous pollen grains remained ungerminated and no pollen tube entering the ovary was observed. In this case, mechanisms inhibiting pollen-tube growth probably appeared in the early stages of its growth. The relatively high number of pistils with ungerminated pollen grains of *R. illyricus* (54.5%) could also be a result of low viability of pollen (Table 3).

*Ranunculus illyricus*, which is a rare and endangered species in many countries, deserves special attention. Active conservation measures should take into account the fact that the fruit set of *R. illyricus* is low in ex situ conditions, perhaps owing to geitonogamy. It cannot be ruled out that this phenomenon also affects the wild population, but the biology of *R. illyricus* has not yet been investigated.

## 5. Conclusions

Our comparative analysis of reproductive traits of five *Ranunulus* species and *Ficaria verna* allowed for the distinguishing of four clusters of species, reproducing in sexual (cluster 1), clonal (cluster 2 and 3) and apomictic (cluster 4) modes. The clonal species, which belong to two separate clusters, were characterized by lower efficiency of sexual reproduction compared with the nonclonal species. However, within the nonclonal species, an additional group of apomictic species was separated. The new divisional clusters coincide with classical systematic sections identified within the genus *Ranunculus* by Tutin based on morphological criteria.

The reproductive efficiency, expressed as the development of viable seeds able to germinate, as well as the susceptibility of seeds to dormancy-breaking factors, varied across the species. Gibberellin treatments improved seed germination in contrast to low temperature, which inhibited germination of *R. bulbosus* seeds.

Great potential for sexual reproduction expressed by the number of pistils in *Ranunculus illyricus* flowers was limited by the relatively low viability of pollen and its poor performance in the style. The fruit set efficiency in *R. illyricus* was low, however, seeds were viable. It is important that methods of dormancy breaking should be developed to support future conservation efforts.

## Figures and Tables

**Figure 1 biology-11-00085-f001:**
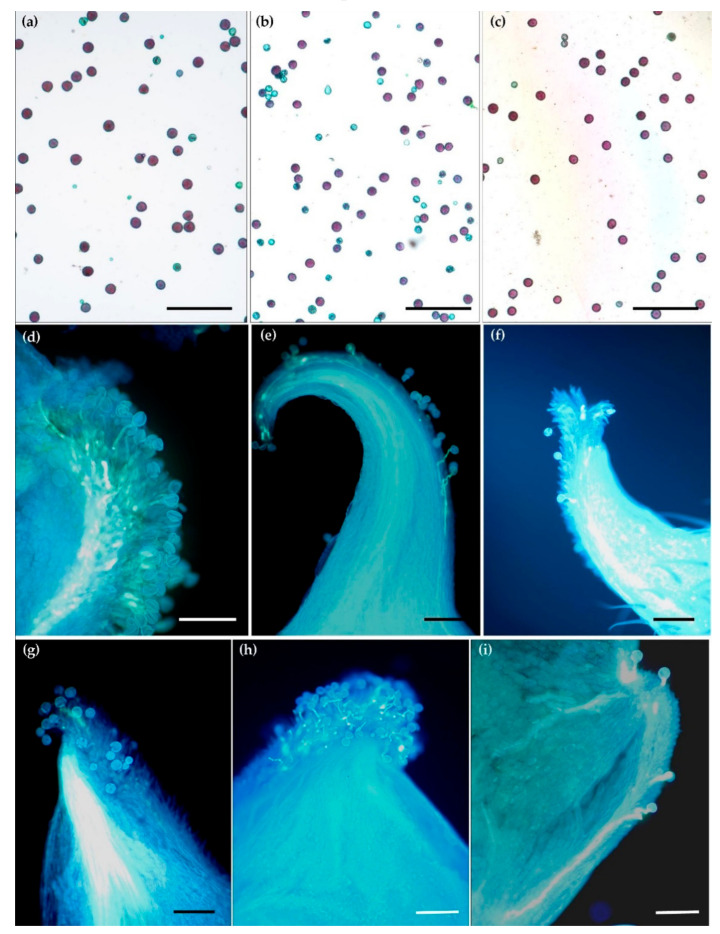
Pollen grains’ viability and germination for *Ranunculus* species. Pollen grains’ viability for (**a**) *R. illyricus*, (**b**) *Ficaria verna*, (**c**) *R. bulbosus* after Alexander method staining: red—alive, green—dead. (**d**–**i**) Pollen grains’ and pollen tubes’ growth in fluorescent ultraviolet light after aniline blue staining; (**d**)—germinating pollen grains on stigma of *F. verna*, (**e**) *R. lanuginosus*, (**f**) *R. auricomus*, (**g**) *R. illyricus*, (**h**,**i**) *R. bulbosus*, bar = 200 μm.

**Figure 2 biology-11-00085-f002:**
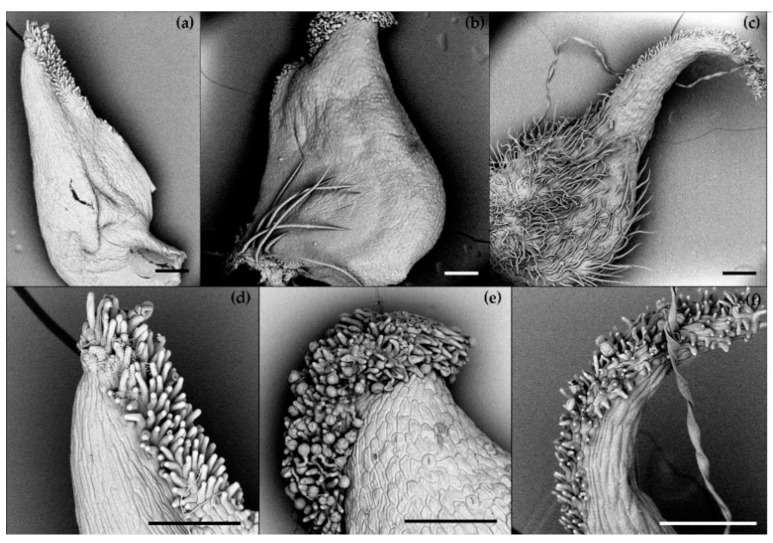
Pistil morphology of *Ranunculus* species, SEM; (**a**,**d**)—*R. illyricus,* (**b**,**e**)—*R. bulbosus*, (**c**,**f**)—*R. cassubicus*; bar = 200 μm.

**Figure 3 biology-11-00085-f003:**
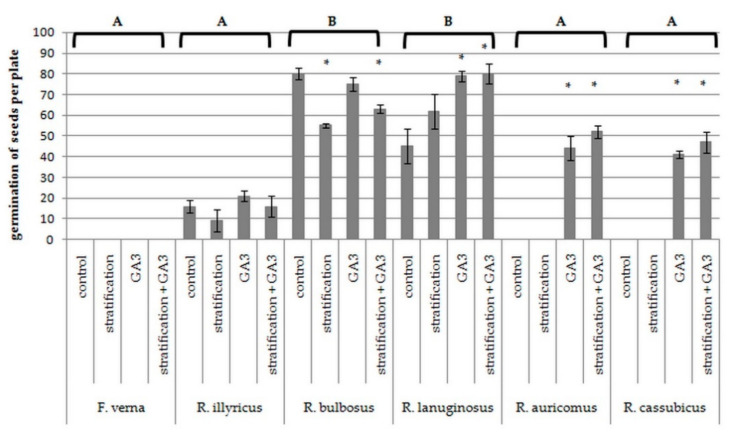
Ability of seeds to germinate after exposure to low temperature and GA_3_ treatment. Letters indicate differences between species according to Kruskal-Wallis and Dunn’s test with α = 0.05. * indicate differences between each treatment and control according to U-Mann–Whitney test with α = 0.05 separately for each species.

**Figure 4 biology-11-00085-f004:**
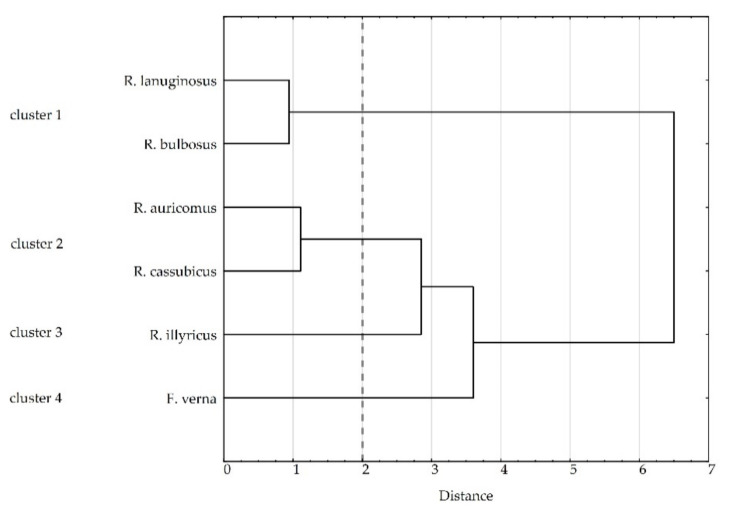
Cluster analysis of species based on pollen viability and diameter, number of pistils per flower, efficiency of fruit set, viability and seed germination (Euclidean distance, Ward’s method). Dashed line marks cut-off point (distance 2) and separation into 4 clusters.

**Table 1 biology-11-00085-t001:** Systematics affiliation, ploidy level, reproduction mode and breeding system of investigated *Ficaria verna* and *Ranunculus* species.

Species	Section ^a^	Ploidy Level and Chromosome Number ^b^	Reproduction Type ^c^	Breeding System ^d^
*Ficaria verna* Huds.	Ficaria	4x = 32	G, Cl	FAl
*Ranunculus illyricus* L.	Ranunculastrum	4x = 32	G, Cl	Al, SI
*Ranunculus bulbosus* L.	Ranunculus	2x = 16	G	Al, SI
*Ranunculus lanuginosus* L.	Ranunculus	4x = 28	G	FAl
*Ranunculus auricomus* L.	Auricomus	4(2,5,6)x = 16–48	G	Al, SI, FAp
*Ranunculus cassubicus* L.	Auricomus	4x = 32	G	FAp

^a^ according to Tutin et al. [6]; ^b^ according to PLADIAS [19]; ^c^ according to Erikson [41], Sarukhán and Harper [12], Troll [42], and Tutin et al. [6]: G—generative, Cl—clonal; ^d^ according to PLADIAS: FAl—facultative allogamy, Al—allogamy, SI—self-incompatibility, FAp—facultative apomixis.

**Table 2 biology-11-00085-t002:** Locality and date of plant material collection of *Ficaria verna* and *Ranunculus* species.

Species	Collection Locality	GPS Coordinates	Collection Date	
			Flowers	Fruits
*F. verna*	oak hornbeam forest, Bielany, Kraków (Kraków Gate mezoregion)	50°02′54.6″ N 019°50′15.0” E	21 April 2017 10 April 2019	15 May 2017 1 May 2019
*R. illyricus*	pot cultivation, collection of University of Agriculture in Kraków (Kraków-Częstochowa Upland)	50°05′03.6″ N 019°57′01.3″ E	7 June 2017 10 June 2019	12 July 2017 10 July 2019
*R. bulbosus*	grasslands from *Festuco-Brometea*, Mydlniki, Kraków (Kraków-Częstochowa Upland)	50°05′03.8″ N 019°51′34.7″ E	13 May 2017 20 May 2019	20 June 2017 26 June 2019
*R. lanuginosus*	oak hornbeam forest, Bielany, Kraków (Kraków Gate mezoregion)	50°02′54.6″ N 019°50′15.0″ E	8 May 2017 1 May 2019	22 May 2017 29 May 2019
*R. auricomus*	moist meadow from Molinietalia coeruleae, Wola Radziszowska, Wieliczka Foothills (Western Carpathians)	49°53′51.1″ N 019°46′16.8″ E	19 May 2017 12 May 2019	16 June 2017 6 June 2019
*R. cassubicus*	oak hornbeam forest, Uniejów-Rędziny, Miechów Upland (Małopolska Upland)	50°26′39.5″ N 019°58′43.7″ E	19 May 2017 22 May 2019	10 June 2017 12 June 2019

**Table 3 biology-11-00085-t003:** Traits of generative reproduction of investigated *Ficaria verna* and *Ranunculus* species. Values represent means of parameter ±SE; letters indicate differences between species according to statistical analysis for each parameter, separately. Based on chi-squared analysis, it can be only stated that clusters differ from each other.

Species	Pollen Viability Per Sample (%/Sample)	Pollen Diameter (µm)	Number of Pistils Per Flower (pcs/Flower)	Efficiency of Fruit Set (%)	Viability of Seeds (%)	Germination of Seeds Per Plate (%/Plate)
*F. verna*	84 ± 3.1 bc	37 ± 0.2	12 ± 0.3 a	22 ± 1 b	7	0 ± 0 a
*R. illyricus*	59 ± 2.1 abc	36 ± 0.4	146 ± 3.2 e	11 ± 1.2 a	100	16 ± 2.1 a
*R. bulbosus*	91 ± 2.2 c	36 ± 1.7	32 ± 0.8 c	64 ± 3.2 d	97	68 ± 2.8 b
*R. lanuginosus*	85 ± 2.7 bc	41 ± 0.2	22 ± 0.6 b	51 ± 3.2 cd	67	67 ± 4.7 b
*R. auricomus*	38 ± 1.9 a	35 ± 0.7	49 ± 2.6 cd	35 ± 1.8 c	35	24 ± 6.4 a
*R. cassubicus*	54 ± 2 ab	36 ± 0.5	84 ± 2.8 d	33 ± 1.6 c	48	22 ± 5.8 a
Test	Kruskal–Wallis test	Kruskal–Wallis test	Kruskal–Wallis test	Kruskal–Wallis test	Chi-square	Kruskal–Wallis test
*p*-value	0.0000	0.1036	0.000	0.000	0.00000	0.0000

**Table 4 biology-11-00085-t004:** Pollination effectiveness and pollen-tube elongation in pistil tissues of investigated *Ficaria verna* and *Ranunculus* species.

Species	Pollinated Pistils(%)	Pistils with Nongerminating Pollen	Pistils with Germinating Pollen Not Entering Ovary	Pistils with Pollen Tubes Entering Ovary
			% of Pollinated Pistils	
*F. verna*	100.0	6.5	45.1	48.4
*R. illyricus*	91.0	55.5	54.5	0
*R. bulbosus*	86.2	4	68	28
*R. lanuginosus*	80.0	0	93.7	6.3
*R. auricomus*	78.6	31.8	59.1	9.1
*R. cassubicus*	95.2	15	80	5

**Table 5 biology-11-00085-t005:** Values of traits of generative reproduction in clusters (mean ± SE). Statistical test used is mentioned. Letters indicate differences between clusters according to statistical analysis for each parameter, separately. In the case of chi-squared analysis, it can be only stated that clusters differ from each other.

Parameter	Cluster 1	Cluster 2	Cluster 3	Cluster 4	Test	*p* Value
	*R. lanuginosus* *R. bulbosus*	*R. auricomus* *R. cassubicus*	*R. illyricus*	*F. verna*		
Pollen viability (%)	88 ± 1.9 c	46 ± 2.7 a	59 ± 2.1 b	84 ± 3.1 c	ANOVA, LSD Fisher’s test	0.000000
Number of pistils per flower	27 ± 0.6 b	68 ± 2.5 c	145 ± 3.2 d	12 ± 0.3 a	Kruskal–Wallis test	0.000
Efficiency of fruit set (%)	58 ± 2.3 d	34 ± 1.2 c	11 ± 1.2 a	22 ± 1 b	Kruskal–Wallis test	0.000
Viability of seeds (%)	82	42	100	7	Chi-square	0.00000
Germination of seeds per plate (%)	67 ± 2.7 b	23 ± 4.3 a	16 ± 2.1 a	0 ± 0 a	Kruskal–Wallis test	0.0000
Pollen diameter (µm)	39 ± 1.5 a	36 ± 0.4 a	36 ± 0.4 a	37 ± 0.2 a	Kruskal–Wallis test	0.4439

## Data Availability

The data presented in this study are available on request from the corresponding author.

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
