# Peer review of "The Effectiveness of the Sexual Reproduction in Selected Clonal and Nonclonal Species of the Genus Ranunculus"

_biology, 2022, doi:10.3390/biology11010085_

Round 1
Reviewer 1 Report
In the simple summary (and also in the abstract) it is not clear why "(...), among the non-clonal species, a cluster analysis separated a third group of 18
apomictic species: R. auricomus and R. cassubicus". Complexity observed leads to ...
L58: Please explain a bit more the potential interest of that question "A question worth answering is to what extend the efficiency of sexual
reproduction depends on the genetically determined reproduction mode.".
The importance of this aspect; " Different strategies preventing self-pollination have been described" in the question supra should be discussed.
The methodology and results are quite relevant and well-explained.
Please explain deeply the data observed that suggest the idea of a separate cluster (from L300).
Author Response
Dear Reviewer,
We appreciate the time and effort that you dedicated to providing feedback on our manuscript “The Effectiveness of the Sexual Reproduction in Selected Clonal and Non-Clonal Species of the Genus Ranunculus” and we are grateful for insightful comments and valuable improvements to it. We have carefully incorporated yours suggestions. The changes are highlighted within the manuscript. Please see below a point-to-point response to the specific comments. Line numbers refer to the revised manuscript file with tracked changes.
We hope that you find our responses satisfactory.
In the simple summary (and also in the abstract) it is not clear why "(...), among the non-clonal species, a cluster analysis separated a third group of 18
apomictic species: R. auricomus and R. cassubicus". Complexity observed leads to ...
The Simple summary has been changed to make it more understandable for the public without specific knowledge. Abstract has been provided (line 48-49) with the features, which allowed to separate the apomictic species from the group of reproducing generatively. We hope that the current form meets expectations of the Reviewer.
L58: Please explain a bit more the potential interest of that question "A question worth answering is to what extend the efficiency of sexual
reproduction depends on the genetically determined reproduction mode."
The question here is whether the method of reproduction assigned to species (vegetative or generative) will have an impact on the efficiency of generative reproduction, studied further in the manuscript. As this is clearly related to the overall purpose of the work, in our opinion changes are not necessary at this place. On the other hand, the problem of mechanisms related to pollination (protandry and protogyny), obviously very interesting in this context, has been detailed later but already on the example of species of the genus Ranunculus (lines 87-92).We are aware that on different stages of new offspring development (both vegatative and generative) various regulating mechanism can appear in relation to e.g. pollination, breeding system and fertilization which affect number of progeny. However, even for Ranunculus genus only, the subject is complex and abundant that would deserve separate elaboration.
The importance of this aspect; " Different strategies preventing self-pollination have been described" in the question supra should be discussed.
Appropriate section concerning protandry and protogyny among Ranunculus species has been added. (lines 87-92)
Please explain deeply the data observed that suggest the idea of a separate cluster (from L300).
The features specific for separated group of apomictic species were introduced in the Discussion.
Yours sincerely
Corresponding author in behalf all authors

Reviewer 2 Report
Dear Authors,
this work is of great interest to the scientific community involved in the study of the biology of reproduction in ornamental species. The work is well written and smooth.
Here are some more of my comments:
- the difference between simple summary and abstract is not clear to me. In this section, however, the aspects relating to the topic under study are not taken into consideration and some main results and conclusions of the work are also missing. If the main text was later revised well, this part is not very clear.
- Line 12: add 'or non clonal'
- Line 67: the term "buttercup" appears for the first time here. Please specify this first.
- the captions of tables and figures are not always specific to the table or figure. Please specify better what they contain. In Table 3, for example, some units of measurement are missing. The statistical test is not needed, you can indicate it in subscript. Please review these parts.
Author Response
Dear Reviewer,
We are very grateful for your review which helped us to improve the manuscript “The Effectiveness of the Sexual Reproduction in Selected Clonal and Non-Clonal Species of the Genus Ranunculus”. We are appreciate your that you recognize the manuscript interesting and we are grateful for your time and comments which helped us to improve the quality of the manuscript.
We have carefully incorporated yours suggestions. The changes are highlighted within the manuscript. Please see below a point-to-point response to the comments. We hope that you find our answers and changes in the manuscript satisfactory.
Specific comments:
The difference between simple summary and abstract is not clear to me. In this section, however, the aspects relating to the topic under study are not taken into consideration and some main results and conclusions of the work are also missing. If the main text was later revised well, this part is not very clear.
The Simple summary has been changed to make it more understandable for the public without specific knowledge. Abstract has been provided with the features, which allowed to separate the apomictic species from the group of reproducing generatively. We hope, that the current form meets expectations of the Reviewer.
line 12: Not applicable, because whole Simple summary has been changed
line 77 (previously 67): in the line 70 the term “buttercups” was introduced together with the latin name of the genus Ranunculus.
- the captions of tables and figures are not always specific to the table or figure. Please specify better what they contain. In Table 3, for example, some units of measurement are missing. The statistical test is not needed, you can indicate it in subscript. Please review these parts.
Captions of Table 1, 2, 3, 4 and Figure 4 has been modified to provide more specific information about content.
In the Table 3 missing units were added.
After consideration, due to different statistical methods used, we find that current form of table would be more clear. However this item we leave the Editor to decide.
Yours sincerely
Corresponding author in behalf all authors
